# Screening accuracy and cut-offs of the Polish version of Communication and Symbolic Behavior Scales-Developmental Profile Infant-Toddler Checklist

**Mateusz Sobieski**[1]*, **Anna Kopszak**[2], **Sylwia Wrona**[3], **Maria Magdalena Bujnowska-Fedak**[1]

**1** Department of Family Medicine, Wroclaw Medical University, Wroclaw, Poland, **2** Statistical Analysis Center, Wroclaw Medical University, Wroclaw, Poland, **3** Faculty of Arts and Educational Sciences, University of Silesia in Katowice, Katowice, Poland

* mateusz.sobieski@student.umw.edu.pl

**Data Availability Statement:** Data can be accessed at https://doi.org/10.6084/m9.figshare.26077021.v1.

## Abstract

### Background

The first stage of diagnosing autism spectrum disorders usually involves population screening to detect children at risk. This study aims to assess the predictive convergent validity of the Polish version of the Communication and Symbolic Behavior Scales-Developmental Profile Infant-Toddler Checklist (CSBS-DP ITC) with the Autism Spectrum Rating Scales (ASRS), evaluate its sensitivity and specificity and assess the cut-off points for the possibility of using this questionnaire in population screening among children aged 6 to 24 months.

### Method

The study was conducted among 602 children from the general population who had previously participated in the earlier phase of validation of the questionnaire for Polish conditions. The collected data were statistically processed to calculate the accuracy (i.e. sensitivity, specificity) of the questionnaire.

### Results

In individual age groups, the sensitivity of the questionnaire varies from 0.667 to 0.750, specificity from 0.854 to 0.939, positive predictive value from 0.261 to 0.4 and negative predictive value—from 0.979 to 0.981. Screening accuracy ranges from 0.847 to 0.923 depending on the age group. The adopted cut-off points are 21 points for children aged 9–12 months, 36 for children aged 13–18 months, 39 for children aged 19–24 months. Cut-off points could not be established for children aged 6–8 months. The convergent validity values with the ASRS ranged from -0.28 to -0.431 and were highest in the group of the oldest children.

**Funding:** The study was financed from the researchers' own funds and the internal funds of Wroclaw Medical University - the project is financed from the scientific grant: "Aspects of preventive care, diagnosis and therapy of patients of different ages under the care of a family doctor, including e-health, telemedicine solutions, coordination and analysis of care effectiveness indicators"; Task SIMPLE number: SUB. C290.19.054. The external funders had no role in study design, data collection and analysis, decision to publish, or preparation of the manuscript.

**Competing interests:** The authors have declared that no competing interests exist.

## Conclusions

These results indicate that the Polish version of the CSBS-DP ITC can be used as an effective tool for ASD universal screening.

## Introduction

Autism spectrum disorders (ASD) are neurodevelopmental conditions of undetermined and complex etiology, most often manifested in early childhood, which affect the everyday activities of individuals and are characterized primarily by difficulties in the sphere of communication and interpersonal interactions as well as restricted interests or repetitive behaviors [1]. The prevalence of ASD is not clearly defined—the World Health Organization (WHO) estimates that ASD affects 1 in 160 children worldwide [2]. More recent reports from a systematic review by Zeidan et al. estimate the worldwide prevalence of ASD as 1 in 100 children [3] at the same time indicating significant variability in prevalence depending on the country and research assumptions used in individual studies. According to the only official data from the Polish National Health Fund from 2012, the prevalence rate of ASD in individuals under the age of 18 in Poland is 3.4 cases per 10,000 children [4], but more accurate preliminary data from two Polish provinces estimate that ASD occurs in one in 286 children [5].

The first symptoms of ASD usually appear during early child development—prodromal symptoms may be visible as early as 6 months of age [6, 7]. Later in development (from 14 months to 3 years of age) communication and social behavioral symptoms become apparent [8]. It is also possible that development plateaus after a period when the age-appropriate milestones were achieved or that the skills acquired earlier are lost [9, 10]. It is assumed that a reliable diagnosis of ASD in a child can be made as early as the 2nd–3rd year of age [11]. However, due to delays associated with the diagnostic process (or its omission), this diagnosis is made much later—in a world-wide 2019 meta-analysis, the average age of ASD diagnosis is 60.48 months (range 30.90–234.57) [12].

Early diagnosis of ASD enables the initiation of an early, age-adjusted therapy of developmental delays and difficulties. The younger the child, the better the results of therapy can be achieved in the area of communication and social interaction, cognitive abilities, speech development, or behavior appropriate to the situation, which improves the quality of life of people with ASD, reduces the risk of mental disorders, and significantly reduces the burden of ASD [13–16].

Considering this, the American Academy of Pediatrics (AAP) recommends screening at 18 and 24 months of age as part of primary care during well-child care visits [17]. It appears that the increasing availability of screening significantly lowered the age of ASD diagnosis in the US, with diagnosis before the age of 4 made in 71% of children (2018) compared to 58% in 2014 [18, 19]. On the other hand, a 2016 report from the US Preventive Services Task Force shows insufficient evidence to recommend universal ASD screening [20]. However, there is evidence suggesting that including screening tools in routine medical appointments may result in earlier and more accurate identification of children who need further help compared to relying solely on clinical impressions, which is particularly important when care providers are less experienced in diagnosing ASD [21]. Moreover, the use of public ASD screening may reduce social inequities in terms of the age of diagnosis and access to further therapeutic activities [22, 23]. The conclusions of both reports indicate the need for further research on screening tools and their effectiveness, as well as on the effectiveness of further proceedings after screening [24].

Due to the lack of tools for early diagnosis of ASD available in Poland, the authors have prepared a linguistically and culturally adapted version of the original, American version of Communication and Symbolic Behavior Scales-Developmental Profile—Infant-Toddler Checklist (CSBS-DP ITC) [25]. CSBS-DP ITC is one of the available tools created for the early detection of symptoms of autism spectrum disorders. It is a 24-item questionnaire for parents or caregivers. The included questions are arranged into three composites (social, speech, and symbolic composite) and seven development predictors (emotion and eye gaze, communication, gestures, sounds, words, understanding, and object use). Original CSBS-DP ITC version can be used in universal ASD screening of children aged 6 to 24 months in a primary care setting [26]. The research showed that the result obtained in the original version of questionnaire could confidently predict the level of language development two years in advance and is an effective tool for the screening of children with special needs [25–27].

In order to conduct a validation study and disseminate knowledge about the research, the "Spojrzeć w oczy" (Pol. "Look into the eyes") project was established. The project aimed to determine the psychometric properties (e.g. validity and reliability, sensitivity and specificity) of the Polish version of the CSBS-DP ITC. The questionnaire was adapted and translated using the back translate method by three independent translators, and the questions included were adjusted to the phonetics of the Polish language and the speech development of Polish children. A detailed description of the preparation of the tool for Polish cultural conditions is included in the earlier publication on CSBS-DP ITC validation [28]. The Polish version of CSBS-DP ITC is included as S1 File. Questionnaire and its English re-translation as S2 File. Data from the earlier stage of the project indicate a very good fit of the one-factor and three-factor models in confirmatory factor analysis. The total score of the Polish version of CSBS-DP ITC demonstrated satisfactory internal consistency, Cronbach's α = .92, and McDonald's ω = .92. The stability of the measurement was confirmed by performing interrater and test-retest reliability analysis, proving perfect and satisfactory level of stability, respectively.

This study aims to assess the predictive convergent validity of the Polish version of the CSBS-DP ITC with the Autism Spectrum Rating Scales (ASRS) questionnaire, used by psychologists for screening children with suspected ASD and those at risk of it, evaluate the sensitivity and specificity of the Polish version of the CSBS-DP ITC and finally assess the cut-off points for the use of the questionnaire in population screening.

## Methods

### Participants

The participants were children aged 30 months and their parents or caregivers who participated in the earlier phase of the project by correctly filling in the Polish version of CSBS-DP ITC questionnaire when children were between 6 and 24 months of age. The condition for inclusion was living in Poland, speaking Polish as one's primary language, and giving informed, written consent to participate in the study. Project recruitment began on October 25, 2020 using advertisements placed in collaborating health care facilities and on social media, and ended on February 18, 2021. During this period, parents completed the Polish version of CSBS-DP ITC screening questionnaire. In addition, as part of the "Spojrzeć w oczy" project, materials on the symptoms, diagnosis and therapy of autism spectrum disorders were made available to parents and health care specialists—both in physical and online form [29]. The research phase lasted from April 28, 2021, to October 12, 2022 –until the last child included in the project turned thirty months of age. As some of the children (N = 34) were awaiting the final diagnosis, their parents were contacted by phone at a later date. By March 30, 2023, a final diagnosis was obtained in all of them, except one (due to diagnostic

difficulties, it was not possible to unambiguously confirm the presence of autism spectrum disorders in that particular case). During and after the data collection, only MS and SW had the opportunity to identify individual study participants—this was to enable the provision of further psychological and pedagogical assistance and further diagnostics.

Invitations to the follow-up were sent to all 1461 parents of children who were included in the first phase of the project. A total of 678 submissions were received back, 76 reports were excluded from the study– 48 of them because parents had not previously participated in Polish CSBS-DP ITC screening, and in 11 cases siblings were assessed in the follow-up instead of the child who was screened, and another 17 were repeated submissions. When attempts were made to contact the other parents via e-mail or telephone (N = 859), as many as 722 of them admitted that they did not complete the further part of the study due to the lack of symptoms in their children and hence—lack of willingness to remain in the study. Finally, 602 children were enrolled in the study, whose parents had fully completed the ASRS questionnaire and follow-up interview, and who had participated in an earlier CSBS-DP ITC screening, which gives a return rate of 41.2%. The number of participants significantly exceeds the minimum sample size to have a confidence level of 95% and a margin of error of 5% in the Polish population (the calculated minimum sample size was 139) [30]. The mean age of the children was 30.21 months (SD = 1.03). The vast majority of questionnaire forms were filled in by the children's mothers (N = 601, 99.83%)–only one of them was filled in by a father. Sample characteristics of the study participants are presented in Table 1. The individual phases of the project and the number of participants at each stage are collected in Fig 1.

Approval from the Bioethics Committee of the Wroclaw Medical University was obtained to conduct the study (number KB– 641/2020; the full text of the consent is available as S3 File and its translation—as S4 File). All study participants gave written consent to participate in the study. All procedures were performed in accordance with the 1964 Helsinki Declaration and its later amendments.

## Evaluation instruments

**Autism Spectrum Rating Scales (ASRS).** ASRS is a set of questionnaires, used as an auxiliary tool in the diagnostic process or a screening tool, consisting of full and abbreviated versions of tests for younger children (2–5 years) and older children and adolescents (6–18 years), both for parents and teachers [31]. The ASRS was developed in 2009 and its structure corresponds to the diagnostic criteria of Diagnostic and Statistical Manual of Mental Disorders, Fourth Edition, Text Revision (DSM-IV-TR) and International Statistical Classification of Diseases and Related Health Problems, 10th Revision (ICD-10). The Polish version was prepared in 2016 and is characterized by high reliability in the version for parents and has a confirmed discriminatory, convergent, and differential validity [32]. Additionally, the Polish version corresponds to of Diagnostic and Statistical Manual of Mental Disorders, Fifth Edition (DSM-5) diagnostic criteria. In the study, the short version for younger children was used, which contains 15 items that best differentiate between children diagnosed with ASD and the comparative group from the non-clinical population. The analysis of the discriminant function for raw scores revealed that the indicators of correct classification ranged from 88.2% to 91.4%. Cronbach's alpha for the ASRS version used in the study is 0.85. Sensitivity and specificity are 87.8% and 83.5%, respectively.

**Further evaluation instruments.** As ASRS is a Level 2 ASD diagnostic tool, parents were asked for additional information to determine the need for further diagnosis. In order to further evaluate children's development, a short structured interview (follow-up interview) was created. They were also asked if their child had received an ASD diagnosis from a psychiatrist

**Table 1. Sociodemographic characteristics of participants.**

| Characteristic | Follow-up | |
|---|---|---|
| | *n* | *%* |
| Sex | | |
| Female | 248 | 41.20 |
| Male | 354 | 58.80 |
| Preterm born | | |
| >34 weeks | 1 | 0.16 |
| 34–37 weeks | 17 | 2.82 |
| Medical conditions | | |
| Serious genetic disorders | 4 | 0.66 |
| Serious health problems* | 24 | 3.99 |
| Sight problems | 5 | 0.83 |
| Hearing problems | 4 | 0.66 |
| Muscle tone or other musculoskeletal system disorders | 118 | 19.60 |
| Physical rehabilitation in the past | 240 | 39.86 |
| ASD in closest family (first degree relatives) | 7 | 1.16 |
| Place of residence | | |
| Village | 140 | 23.25 |
| Town inhabited by less than 20,000 people | 54 | 8.97 |
| City inhabited by 20,000–100,000 people | 92 | 15.28 |
| City inhabited by more than 100.000 people | 316 | 52.49 |

**Note**.

*We asked parents to list comorbidities that seemed important to them as "serious health problems". At a later stage, we verified the answers, taking into account only those that, in our opinion, may have any impact on the child's development (examples of disorders reported by parents include neurological diseases, heart and kidney malformations, phenylketonuria, perinatal disorders, i.e. hypoxia, congenital adrenal hyperplasia, neuroblastoma, esophageal atresia, etc.)

(understood as disorders included in the ICD-10 classification as F84.0, F84.1, or F84.5 or in the DSM-5 as autism spectrum disorder). They were also asked if their child had completed any of the available standardized diagnostic protocols available in Poland: Autism Diagnostic Interview-Revised (ADI-R), Psychoeducational Profile-3rd Edition-PL (PEP-3-PL), or Autism Diagnostic Observation Schedule, 2nd Version (ADOS-2) and what was the result of the conducted examination [33–35]. Information on the diagnostic provider's credentials and experience was collected when possible to avoid the possible impact of extraneous variables on the results. A positive response in terms of receiving a nosological diagnosis of ASD from a psychiatrist or a positive ADOS-2 test result meant qualification to the group of children diagnosed with ASD. If the child received another diagnosis (e.g. language delay—LD), this was also recorded in the database used in the study.

## Procedures

Due to the COVID-19 pandemic and difficulties in conducting such a study face-to-face in healthcare clinics, an electronic version of the questionnaire was prepared and made available on the project's website examining the properties of the tool. Initially, the parents completed the Polish Version of CSBS-DP ITC questionnaire along with a short record of the mother's and father's age and the child's comorbidities (genetic, hearing and vision disorders,

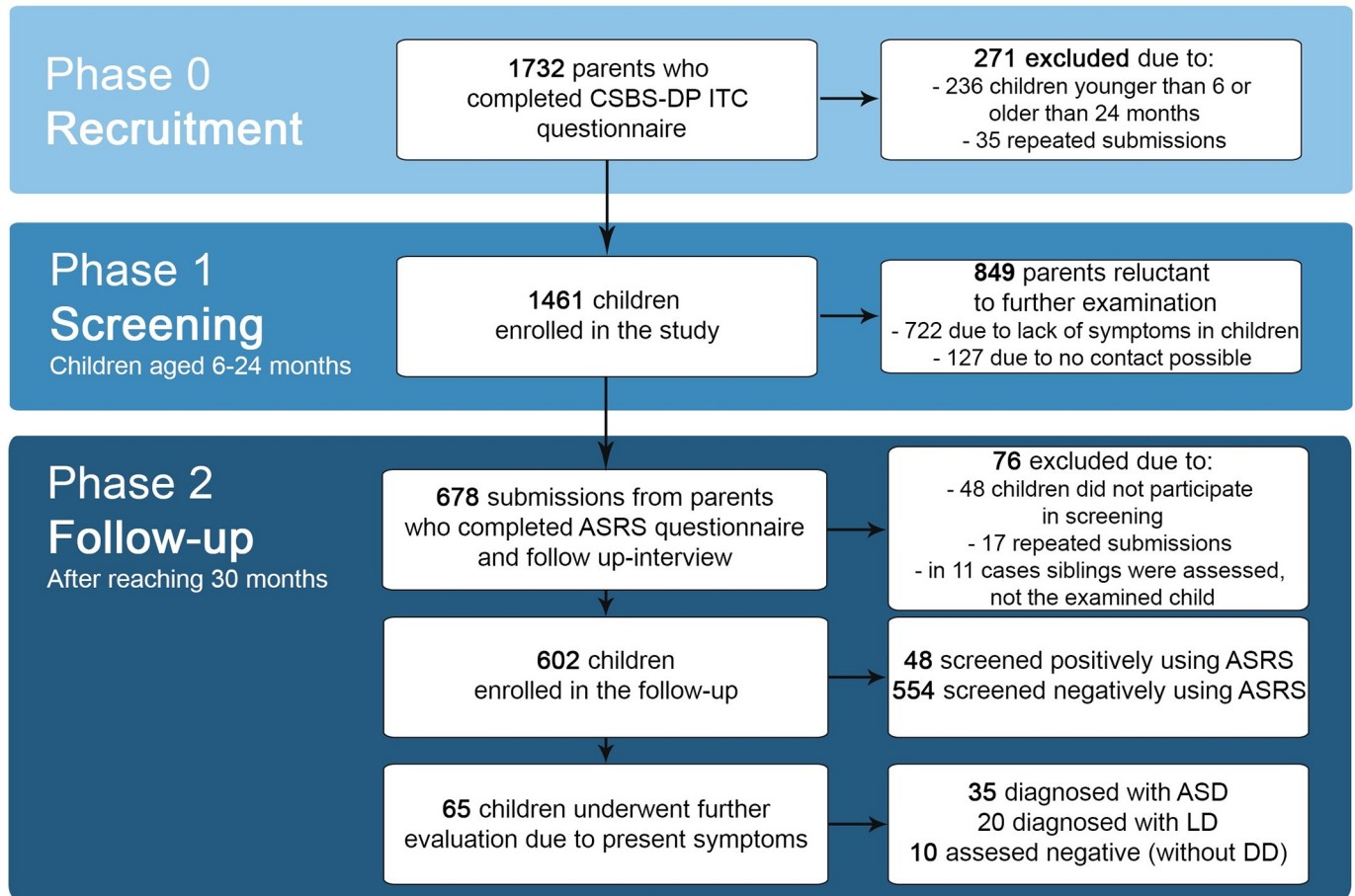

**Fig 1. Chart showing the number of study participants at each stage of the Spojrzeć w oczy project. Note**. ASRS—Autism Spectrum Rating Scales, CSBS-DP ITC—Communication and Symbolic Behavior Scales-Developmental Profile—Infant-Toddler Checklist, Polish Version, DD—developmental delay, LD—language delay.

movement disorders, family history), birth weight, and the week of pregnancy in which the delivery took place. Then, upon a child's turning 30 months of age, the respective child's parents were contacted again and asked to fill in the ASRS questionnaire and a short follow-up interview, which included questions about the occurrence of symptoms that were concerning to the parents and further diagnostics.

In the case of suspicion of ASD in a child (understood as a result above ASRS cut-off point or any persistent concerns parents have about their child's development despite negative screening using the ASRS), the ADOS-2 was performed to make it possible to determine with high probability the presence of autism spectrum disorders in children from the risk group. At each stage during the study, parents were offered the possibility of carrying out the ADOS-2 test at the research center in Cieszyn, Silesian Voivodeship. However, due to the distribution of the population throughout the country, the majority of them preferred performing further diagnostics near their place of residence. If children underwent the ADOS test before our intervention or the ADOS test was performed in another center, we tried to verify the test by contacting the examiner to confirm whether the test result indicated the presence of autism spectrum disorders in the child.

**Statistical analysis.** All the analyses presented in this manuscript were performed using Statistica 13.1 and R 4.3.0 software using packages *ggplot2 3.5.1* and *stats*. Due to the wide variation of results in individual age groups (resulting from rapid development in the age group analyzed) and the insufficient number of participants at a particular age in months, it was necessary to group the participants into age cohorts gathering children at least partially at a similar level of psychomotor development. Participants were divided into four groups based on the substages of Piaget's Sensorimotor Stage theory of cognitive development [36]. The first group included children aged from 6 to 8 months of age, the second group included children between 9 and 12 months, the third group included children between 13 and 18 months, and the fourth group included children above 19 months.

Because more complex tools are also used during further diagnostics of children with suspicion of ASD, first it was checked whether there was any correlation between the results achieved by children in the CSBS-DP ITC questionnaire and the shortened version of the ASRS (used as a Level 2 questionnaire) using the Spearman's rank correlation coefficient method. In addition, since the children were grouped into four age cohorts, it was checked whether the shape of the relationship between CSBS-DP ITC Total Score and Total ASRS is influenced by the variable "Age". The last aspect that was checked was the relationship between the variables of the child's age (in months) and the Total Score obtained in the CSBS-DP ITC questionnaire. To verify the linearity, a normality analysis of the residuals of the linear model was performed using the Kolmogorov—Smirnov and Shapiro—Wilk tests. The tests showed no normality of the residuals in the model. For this reason, non-parametric methods were used for further analysis.

The discriminating ability of CSBS-DP ITC was examined to ensure that it was appropriate for each age cohort. This was achieved by conducting a receiver operating characteristics (ROC) analysis for all four age groups. The discriminating ability of the Polish Version of CSBS-DP ITC was determined by the Area Under the Curve (AUC) statistic. AUC values ranged from 0.5, representing a random chance efficacy, to 1.0, representing excellent performance in discriminating between the two conditions [37]. For each of the groups for which the ROC analysis was successful, potential cut-offs were determined to enable the tool to be used in practice as a clinical guide for further management. Two different methods were used to determine the cut-offs—the first was the cut-off point at which maximum sensitivity and specificity were determined using the Youden index, which represents the overall accuracy of the test [38, 39]. Additionally, the minimizing expected costs method was used by including a "decision threshold" in constructing tangents because potentially classifying a child with ASD as a healthy individual carries a higher cost (in terms of delay in treatment) than the opposite mistake (Zweig & Campbell, 1993). We assumed that the weight of these errors (misclassification costs) increases with the age of the child (and therefore with the delay in diagnosis). It was further assumed that the costs of misclassification (for children falsely classified as healthy—false negative) would be measured as 1, 2, 3, and 5, for the respective groups as age increases; values were adopted arbitrarily.

Due to the unclear prevalence of ASD in the Polish population, three different probabilities were used to determine the cut-off points using the tangent method—random (0.5), declared by WHO for the world population (1:160) and the probability estimated from the study population (depending on the number of cases in a given cohort). Thus, a total of twelve ROC curves were created and analyzed, and the results of these analyses were used to establish the sensitivity and specificity in specific age groups.

**Table 2. Sex structure and incidence of developmental disorders in the studied age subgroups.**

| Variables | Group I (6–8 months of age) N = 73 | Group II (9–12 months of age) N = 131 | Group III (13–18 months of age) N = 242 | Group IV (19–24 months of age) N = 156 |
|---|---|---|---|---|
| Sex | | | | |
| Male | 42 (57.53%) | 75 (57.25%) | 154 (63.64%) | 83 (53.21%) |
| Female | 31 (42.47%) | 56 (42.75%) | 88 (36.36%) | 73 (46.79%) |
| Presence of DD | | | | |
| ASD | 2 (2.74%) | 8 (6.11%) | 16 (6.61%) | 9 (5.77%) |
| LD | 4 (5.48%) | 6 (4.58%) | 7 (2.89%) | 3 (1.92%) |

**Note**. ASD—autism spectrum disorders, DD—developmental disorders, LD—language delay.

## Results

### Descriptive statistics

In the study population, among 602 children, 65 of them were further evaluated due to a positive ASRS test result or the presence of disturbing symptoms. Ultimately, 35 children from the entire group were diagnosed with ASD and 20 with language development delay. Data on the sex structure and incidence of developmental disorders in individual age-subgroups are detailed in Table 2.

The largest number of ASD cases (both percentage and total) were confirmed in the subgroup of children aged 13–18 months (n = 16; 6.61%), the smallest—in the subgroup of children aged 6 to 8 months (n = 2; 2.74%). In each age group, boys slightly dominated the study population (from 53.21% to 63.64%).

### Correlation between total scores of ASRS and CSBS-DP ITC

To assess predictive accuracy between the earlier CSBS-DP ITC questionnaire result and the later ASRS, Spearman's rank correlation coefficient analysis was used. Since the Total Score achieved in the CSBS-DP ITC questionnaire depends on the age of the examined child, it was additionally checked whether the child's age affects the strength of the relationship between Total CSBS-DP ITC scores. The results are presented in the Table 3 and Figs 2 and 3.

Statistically significant weak or moderate correlations are present between the total ASRS score and the total CSBS-DP ITC score in the three oldest age groups. It should be noted that in each age cohort, there are weak or moderate correlations between the child's age and the total score obtained in CSBS-DP ITC. It should be borne in mind that these two questionnaires have an opposite scale—in the CSBS-DP ITC, the higher the score, the lower the risk of ASD in the child, while for the ASRS the opposite is true, which is clearly visible in Fig 2. The older the children examined, the more pronounced the inverse relationships between the results achieved in the CSBS-DP ITC questionnaire and the ASRS. In order to better present this relationship, an analysis was performed for age groups using ranking (due to the previously mentioned lack of normal distribution of variables).

### Sensitivity, specificity of the Polish version of CSBS-DP ITC

In order to determine the sensitivity, specificity and other measures of the classification test for the Polish version of the CSBS-DP ITC, this study used the Youden method of ROC analysis and the tangential method to minimize the potential costs associated with misclassification.

**Table 3. Spearman correlation values for the total scores obtained by children in the appropriate age groups in the ASRS, CSBS-DP ITC questionnaire and age when the CSBS-DP ITC was completed (in months).**

| Variables | Spearman's r | t | p |
|---|---|---|---|
| I (6–8 months of age; N = 73) | | | |
| Total ASRS score & Total CSBS-DP ITC score | -0.118 | -1.000 | 0.321 |
| Total CSBS-DP ITC score & Age of child | **0.389** | **3.555** | **0.001** |
| II (9–12 months of age; N = 131) | | | |
| Total ASRS score & Total CSBS-DP ITC score | **-0.280** | **-3.315** | **0.001** |
| Total CSBS-DP ITC score & Age of child | **0.510** | **6.739** | **<0.001** |
| III (13–18 months of age; N = 242) | | | |
| Total ASRS score & Total CSBS-DP ITC score | **-0.364** | **-6.055** | **<0.001** |
| Total CSBS-DP ITC score & Age of child | **0.412** | **7.013** | **<0.001** |
| IV (19–24 months of age; N = 156) | | | |
| Total ASRS score & Total CSBS-DP ITC score | **-0.431** | **-5.921** | **<0.001** |
| Total CSBS-DP ITC score & Age of child | **0.160** | **2.009** | **0.046** |

**Note**. ASRS—Autism Spectrum Rating Scales, CSBS-DP ITC—Communication and Symbolic Behavior Scales Infant-Toddler Checklist

Results of the analyses and ROC charts for individual age groups using the Youden method are provided in Fig 4a–4c and Table 4. Results using the tangential method with different probabilities of ASD occurrence, due to its lower effectiveness compared to Youden's method, are included in SF5.

Due to the small number of ASD cases in the first group (children aged 6 to 8 months), the ROC analysis in this subgroup could not be performed.

The highest sensitivity and specificity values were achieved using the Youden method and cut-offs did not change depending on the assumed probability of ASD occurrence—when using the prevalence "from the sample", random (0.5) and prevalence based on data from

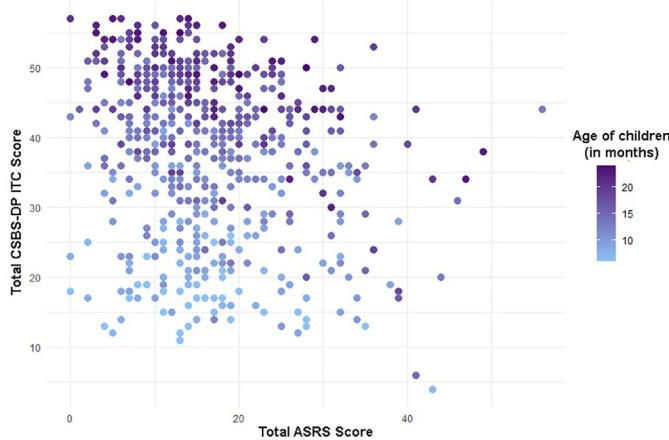

**Fig 2. Distribution of children's total scores in the CSBS-DP ITC and ASRS questionnaires by children's age.** Each dot indicates one child's score on the CSBS-DP ITC questionnaire (Y-axis) in relation to the score on the ASRS questionnaire (X-axis), the color corresponds to the child's age at the time of completing the CSBS-DP ITC questionnaire. Lower scores on the CSBS-DP ITC are also observed in the group achieving low ASRS scores due to lower age.

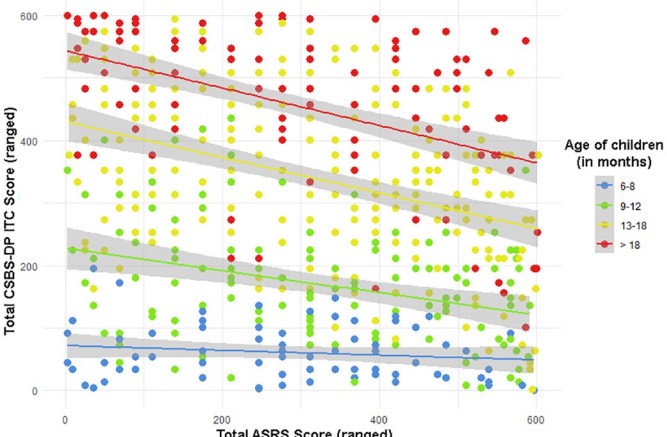

**Fig 3. Rank analysis of the distribution of children's total scores achieved in the CSBS-DP ITC and ASRS questionnaires in the accepted age groups.** Each dot indicates the rank score one child's score on the CSBS-DP ITC questionnaire (Y-axis) and ASRS (X-axis) with a marked trend (which is the highest for the oldest children and the lowest—for the youngest).

systematic reviews (1:160 according to WHO), the ROC curves plotted using the Youden method did not differ from each other. In the case of using the tangential method, the adopted cut-offs significantly differed depending on the assumed probability of ASD occurrence in the Polish population and they were usually characterized by lower accuracy or a significant disproportion between sensitivity and specificity in favor of one of these parameters. Moreover, adopting cut-offs determined using this method may be too volatile (e.g., 30 or 54 points in group 19–24 months of age), which is also probably due to the small number of cases of children diagnosed with ASD in the sample.

Among all performed ROC analyses, the area under the curve (AUC) index ranged from 0.782 to 0.856, which is a value allowing for the classification ability for the CSBS-DP ITC to be considered as at least above moderate. Among children in the age group of 9–12 months, a cut-off of 21 points was adopted, allowing for a sensitivity of 0.750 and a specificity of 0.862 with a Youden index of 0.612; in the age group of 13–18 months, a cut-off of 36 points was adopted, giving a sensitivity of 0.750 and a specificity of 0.854 with a Youden index of 0.604, and in the group of 19–24 months—a cut-off of 39 points, giving a sensitivity of 0.667 and specificity of 0.939 with a Youden index of 0.605.

Due to the size of the groups, analyzes of the CSBS-DP ITC subscales (Social, Speech, Symbolic components) using ROC curves did not allow for clear cut-off points for children. For this reason, if it is necessary to use subscales to determine a child's risk, we suggest using the method used in the original version of the CSBS-DP ITC questionnaire (i.e. 1.25 SD below the mean score in the study population). Raw results and cut-off thresholds for this method were described in a previous publication on the Polish version of CSBS-DP ITC [28].

## Discussion

The aim of this study was to determine the final psychometric values of the Polish version of the CSBS-DP ITC questionnaire as part of the nationwide project. The questionnaire enables testing children for developmental disorders from the age of 6 months, which is a distinctive feature of this study—previous studies on other diagnostic tools e.g. Modified-Checklist for Autism in Toddlers (Revised) (M-CHAT-R/F), Brief Infant Toddler Social Emotional

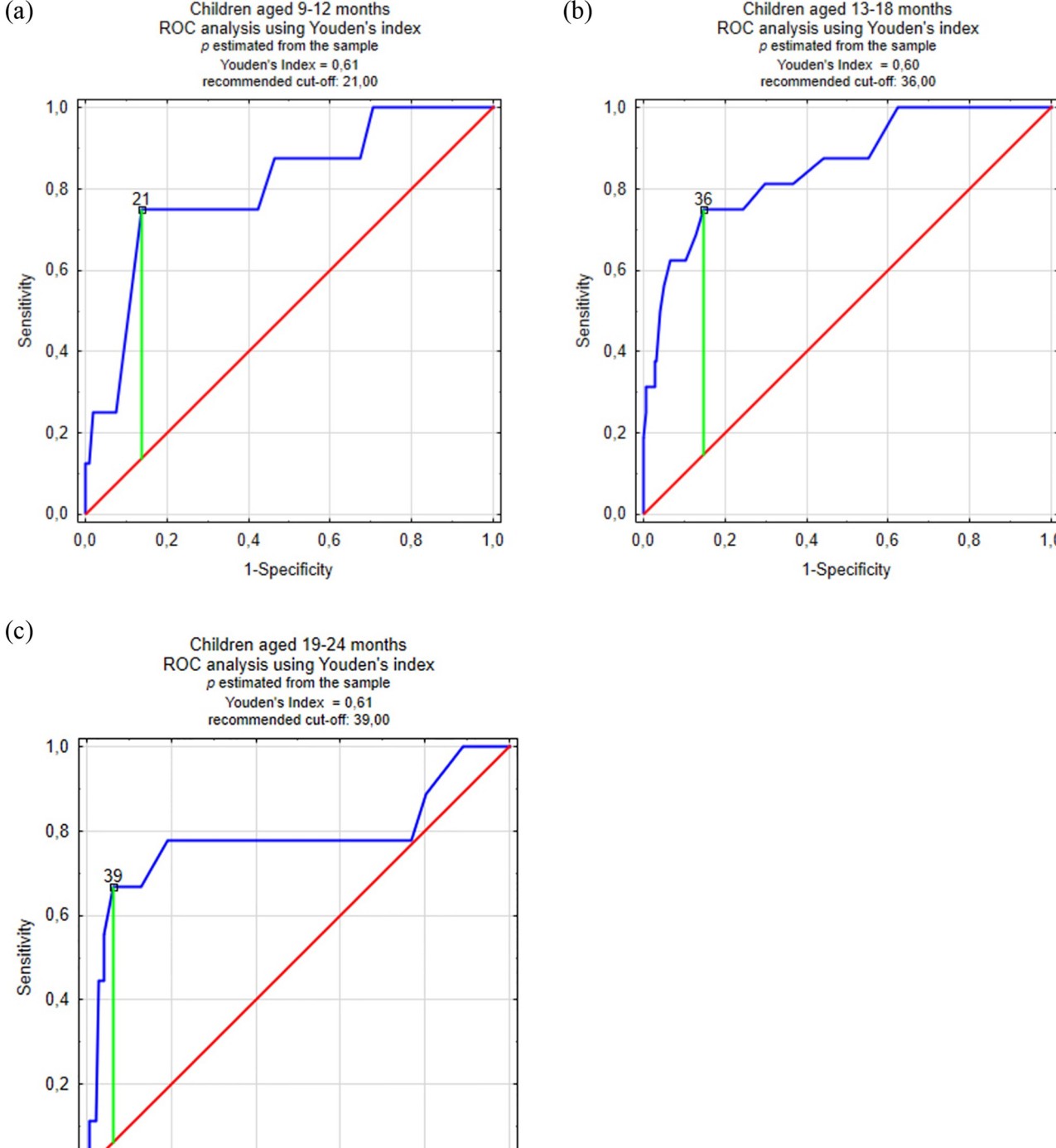

**Fig 4. A-C.** ROC analysis charts in individual age groups using the Youden method.

**Table 4. Values of sensitivity, specificity and other parameters for adopted, best matching cut-off points in given age groups.**

| Total result (Cut-off) | Sensitivity | Specificity | Youden index | Accuracy | PPV | NPV | FPR | FNR | LR(+) | LR (-) |
|---|---|---|---|---|---|---|---|---|---|---|
| *Group II (children aged 9–12 months; N = 131)* *AUC = 0.804; z = 3.664; p < 0.001* | | | | | | | | | | |
| 19 | 0.375 | 0.911 | 0.286 | 0.878 | 0.214 | 0.957 | 0.089 | 0.625 | 4.193 | 0.686 |
| 20 | 0.500 | 0.894 | 0.394 | 0.870 | 0.235 | 0.965 | 0.106 | 0.500 | 4.731 | 0.559 |
| **21** | **0.750** | **0.862** | **0.612** | **0.855** | **0.261** | **0.981** | **0.138** | **0.250** | **5.426** | **0.290** |
| 22 | 0.750 | 0.821 | 0.571 | 0.817 | 0.214 | 0.981 | 0.179 | 0.250 | 4.193 | 0.304 |
| 23 | 0.750 | 0.780 | 0.530 | 0.779 | 0.182 | 0.980 | 0.220 | 0.250 | 3.417 | 0.320 |
| *Group III (children aged 13–18 months; N = 242)* *AUC = 0.856; z = 6.941; p < 0.001* | | | | | | | | | | |
| 34 | 0.625 | 0.898 | 0.523 | 0.880 | 0.303 | 0.971 | 0.102 | 0.375 | 6.141 | 0.417 |
| 35 | 0.688 | 0.872 | 0.559 | 0.860 | 0.275 | 0.975 | 0.128 | 0.313 | 5.358 | 0.359 |
| **36** | **0.750** | **0.854** | **0.604** | **0.847** | **0.267** | **0.980** | **0.146** | **0.250** | **5.136** | **0.293** |
| 37 | 0.750 | 0.823 | 0.573 | 0.818 | 0.231 | 0.979 | 0.177 | 0.250 | 4.238 | 0.304 |
| 38 | 0.750 | 0.757 | 0.507 | 0.756 | 0.179 | 0.977 | 0.243 | 0.250 | 3.082 | 0.330 |
| *Group IV (children aged 19–24 months; N = 156)* *AUC = 0.782; z = 2.597; p < 0.001* | | | | | | | | | | |
| 35 | 0.444 | 0.959 | 0.404 | 0.929 | 0.400 | 0.966 | 0.041 | 0.556 | 10.889 | 0.579 |
| 38 | 0.556 | 0.959 | 0.515 | 0.936 | 0.455 | 0.972 | 0.041 | 0.444 | 13.611 | 0.463 |
| **39** | **0.667** | **0.939** | **0.605** | **0.923** | **0.400** | **0.979** | **0.061** | **0.333** | **10.889** | **0.355** |
| 40 | 0.667 | 0.925 | 0.592 | 0.910 | 0.353 | 0.978 | 0.075 | 0.333 | 8.909 | 0.360 |
| 41 | 0.667 | 0.918 | 0.585 | 0.904 | 0.333 | 0.978 | 0.082 | 0.333 | 8.167 | 0.363 |

Note. Bold values correspond to the cut-off value that has the best predictive parameters according to the ROC analysis using the Youden method. Sensitivity—true positive rate; specificity—true negative rate; PPV—positive predictive value (presicion); NPV—negative predictive value; FPR—false positive rate (fall-out rate); FNR—false negative rate (miss rate); LR(+)–positive likelihood ratio; LR(-)–negative likelihood ratio.

Assessment (BITSEA), First Year Inventory (FYI) or Quantitative Checklist for Autism in Toddlers (Q-CHAT) were focused on older children [40–43]. In addition to the CSBS-DP ITC, the only screening questionnaire designed to be completed by parents (rather than trained observers) for children aged 6 months and older is the Canadian-validated Autism Parent Screen for Infants (APSI) [44]. Therefore, this study may provide evidence for the purposefulness of screening for developmental disorders among the youngest children in the primary care setting.

Due to the dynamic development of children's skills in the studied period, it was necessary to take into account the age difference affecting the results achieved by children. In the original version of the CSBS-DP ITC questionnaire, there are separate cut-off points for each individual month of age, however, they have been set as a point below 1.25 SD from the average result in the general population. Due to the lack of sufficient participants in the study with a confirmed diagnosis of ASD, only solution to this problem was to group children into appropriate larger age cohorts.

Data from the analysis of the statistical model indicate that the correlation between the child's age and the total score in the CSBS-DP ITC questionnaire is also present in age subgroups, which means that children close to the age limit of the cohort may be classified differently by the questionnaire. The influence of age on the total score would probably be even stronger if it was examined without grouping children into cohorts, however, due to its significance, this effect cannot be ignored. Moreover, the analyses do not confirm the linearity of this impact—the introduction of a simple correction based on a linear relationship would involve

the risk of error. Therefore, this is another reason why it was necessary to use previously developed theoretical constructs (in this case, Piaget's theory) to divide children into larger age groups. This, in turn, facilitated the conduct of analyses and the determination of threshold values according to the observed acquisition of particular abilities by children at the appropriate age. With the amount of data available, unfortunately, the only remaining solution is to take into account the correlation with age when interpreting the results and leave a margin of distrust for the scale. This is of particular relevance when evaluating children on the verge of age ranges. In case of doubt, as suggested by the authors of the original version, in the current study it is also recommended to check the child's development in the next three months.

Based on a sample of 1,461 simple screening questionnaires, at least 35 children were diagnosed with ASD, which gives a prevalence of ASD in the Polish population of 2.39%–considering that the majority of parents who did not participate in the follow-up with the use of ASRS and follow-up interview dropped out due to the lack of suspicion of any symptoms of developmental disorders in their children. This percentage is even higher if the study considers only children assessed using the ASRS and subsequent tests (e.g. ADOS-2) and it may be estimated as 5.81%. Compared to the preliminary data from two Polish regions (which indicate 32–38 cases of ASD per 10,000 people), the prevalence of ASD in the group of respondents in the current study is markedly higher. The percentage of children diagnosed with ASD in whole project (2.39%) is therefore similar to the most recent data for American children (1:64) [19]. Similarly, the potentially unusual high percentage of children who underwent rehabilitation or with detected muscle tension disorders and constituted the study group does not seem to differ significantly from the general Polish population. This results from the high frequency of their detection in the Polish population in accordance with the guidelines of the Children and Youth Rehabilitation Section of the Polish Rehabilitation Society, as well as the high popularity of rehabilitation in Poland [29]. According to data collected on behalf of the National Chamber of Physiotherapists, as many as 49.9% of Poles use the services of physiotherapists; however, current accurate data on children are lacking [30].

However, it should be borne in mind that the selection of the method (due to the COVID-19 pandemic, the study was conducted online) may have resulted in some bias of the group—the questionnaire was filled in by willing parents, so parents observing the occurrence of any deficits in their children may account for a larger part of respondents than it actually could be the case in stationary primary care setting. Nevertheless, every effort was made to convince also parents who do not suspect any developmental disorders in their children to complete the questionnaire. It seems that these efforts were effective– 383 parents from 1461 included in the study reported concerns about their child's development, which constitutes 25.7% of all participants. This is perfectly in line with the data from a study conducted at the C.S. Mott Children's Hospital, Michigan, US, where it was shown that a quarter of parents also report suspected developmental disorders in their children [45]. Similar conclusions can be drawn from studies in the Netherlands, where even up to 50% of parents reported minor concerns about their children's development [46]. Notwithstanding the foregoing, the data from the current study indicate that despite the potential bias of the group, there is still a large proportion of undiagnosed individuals in the general population who could benefit from appropriate diagnosis and treatment.

In order to reduce the rate of false positive results, it may be effective to include an observational study as a follow-up measure immediately after a positive result. The authors of the original version of the CSBS-DP created additional, further tools that aim to increase the sensitivity and specificity of the ITC questionnaire (i.e. Behavior Sample, where a pre-trained healthcare professional (HCP) assesses the child's development, or Caregiver Questionnaire, where the child's development is assessed by e.g. a caregiver in a nursery or nanny) [47].

Numerous evidence has also been described that the inclusion of follow-up methods increases the sensitivity and specificity also in the case of other ASD screening tools (e.g. M-CHAT R/F) [48]. In addition, the use of a Behavior Sample by HCPs immediately after an ITC assessment could reduce the potential dropout and be used as an immediate validation strategy. The use of a follow-up method right after the questionnaire-based screening method could be all the more important because some of the families, even after obtaining a result of the screening tests suggesting the possibility of ASD in the child and indicating the need for further diagnostics, still do not feel concerned about the development of their child as a study performed in Flanders suggests [49]. Major obstacles in trying to incorporate additional diagnostic methods in primary care settings are the excessive workload, insufficient time, and inadequate knowledge of HCPs about ASD [50–52].

Another issue worth mentioning is the possibility of using the CSBS-DP ITC questionnaire to detect developmental disorders other than ASD, e.g. language development delay (LD), which were observed in 20 respondents in our study (3.32%), or to assess strengths or weaknesses in a child's developmental skills. Early detection of children with LD, especially those from high-risk groups, may benefit them in connection with the initiation of appropriate therapy, however, the effects of such therapy are less pronounced than in the case of ASD, as indicated by the results of the Danish SPELL longitudinal study conducted in the general population [53, 54]. The presence of three main CSBS-DP ITC subscales concerning social skills, symbolic skills, and speech development indicates the child's potential resources and the most important deficits and may serve as a cue for early intervention therapists before further, more specialized diagnostics. However, the unequivocal use of this information in practice requires further research on the CSBS-DP ITC questionnaire specifically in risk groups.

Nevertheless, the evidence from this study may contribute to the discussion regarding the inclusion of ASD screening during well-child care visits. So far, no fully validated questionnaire for the diagnosis of ASD in children under 2 years of age has existed in Poland; only one study on the reliability of the Q-CHAT questionnaire was published, without an attempt to estimate the cut-off in the Polish population or assess the sensitivity and specificity of the tool [55]. Taking into account the positive data from the current study on the CSBS-DP ITC properties, the use of this tool in everyday practice of HCPs would seem justified.

Increasing early diagnostic capabilities through the use of CSBS-DP ITC in screening has the potential to provide large population benefits similar to those observed in the United States, associated with a significant reduction in the average age of diagnosis and an increase in the percentage of children diagnosed with ASD [19]. Taking into account the previously mentioned low official prevalence of ASD in Poland (which is most likely due to incomplete diagnosis, difficulties with access to specialists and lack of guidelines on how to deal with a child with suspected ASD), as well as the high average age at which the diagnosis of ASD in children is made implementing of ASD screening could improve the situation of children and their families [29]. It is estimated that the average age of receiving an ASD diagnosis in Poland is 7 years and 3 months, which is much higher than the world average, which worsens the potential outcomes of ASD therapy in terms of communication or language skills [12, 56].

The Polish version of CSBS-DP ITC is characterized by psychometric parameters slightly lower than the original version, and similar to the Taiwanese version (sensitivity 0.77) or the Italian version (maximum sensitivity 0.67, specificity 0.98, PPV 0.6, NPV 0.98); however, the Italian version achieved significantly lower diagnostic values in younger children [26, 57, 58]. The Polish version of the CSBS-DP ITC questionnaire, probably due to linguistic and cultural adjustment, does not have such a large impact of low age on psychometric values, which means that it can be used in screening children from 9 months of age [59]. This makes it

possible to implement early intervention at a very early stage of the child's development, which will probably enable achieving better final effects of tailored therapy [60].

There are several limitations to this study. The first is the relatively small number of study participants whose development was monitored throughout the study. Although this number fully meets the criteria of scientific research, a larger sample of respondents could provide better evidence of the usefulness of ASD screening in the Polish population and, at the same time, better estimate the prevalence of ASD in this age group in Poland. Another limitation is the method of electronic screening and the related self-selection of participants. It is believed to be one of the main factors for unrepresentativeness in studies conducted via online questionnaires. The attitudes of study participants may have influenced their decision to take part in the survey, which is why parents who suspected any developmental disorders in their children were more willing to participate in the study [61]. However, as mentioned earlier in the discussion, the present authors made every effort to avoid self-selection bias and, given the percentage of parents with any concerns about their child's development compared to other inpatient studies, most likely the attempt to make it negligible have been successful.

Another possible limitation is the tracking of children's development using online and telephone methods. Due to remote contact with parents, the authors relied on answers in questionnaires, surveys, and medical documentation provided. Nevertheless, in order to qualify the examined child to the group of children diagnosed with ASD, it was required to present appropriate documents prepared by a qualified psychologist or psychiatrist. The last flaw of the study is the time of observation of children—with the endpoint set at 30 months of age, and the possibility of extending this time if the child is undergoing further diagnostics, has received a positive ASRS result or parents have further concerns about the child's development. At the same time, according to the DSM-V definition of ASD, the symptoms of autism spectrum disorders may become fully apparent also in later years of development. Despite providing psychological help to children who require it, some of them may develop symptoms later—when social demands exceed the limited capacities of the child [1, 62].

## Conclusions

The Polish version of the CSBS-DP ITC questionnaire is characterized by reasonably high values of sensitivity, specificity, and accuracy in children aged from 9 to 24 months and children's performance in the questionnaire may also be a predictor of outcomes in later examinations (i.e. ASRS). The conducted analyses indicate the potential usefulness of the Polish version of the CSBS-DP ITC questionnaire in the everyday practice of healthcare professionals for population screening for ASD. Further research is essential to more accurately define cut-offs among children who are on the verge of the accepted age ranges and among the youngest population (aged 6–8 months).

## Supporting information

**S1 File. Questionnaire—The Polish version of CSBS DP-ITC.**
(PDF)

**S2 File. Questionnaire—English translation of the Polish version of CSBS DP-ITC.** Re-translated (from Polish to English) Polish version of CSBS-DP ITC.
(PDF)

**S3 File. Ethics committee approval.** Full text of Ethics Committee Approval for conducting this study obtained by the Wroclaw Medical University Ethics Committee.
(PDF)

**S4 File. Translated ethics committee approval.** Translated from Polish full text of Ethics Committee Approval for conducting this study obtained by the Wroclaw Medical University Ethics Committee.
(DOCX)

**S5 File. Results and discussion on sensitivity, specificity of the Polish version of CSBS-DP ITC calculated using tangential method.**
(DOCX)

## Acknowledgments

Due to obtaining consent from the authors of the original version of the CSBS-DP ITC questionnaire and the general free availability of the questionnaire, the Polish version of the CSBS-DP ITC questionnaire is covered by the CC BY license, with particular emphasis on the original authors and Paul H. Brookes Publishing Co.

The researchers would like to thank Róża Hajkuś and Kamila and Tomasz Chadaj for their help in recruiting study participants.

## Author Contributions

**Conceptualization:** Mateusz Sobieski, Sylwia Wrona, Maria Magdalena Bujnowska-Fedak.

**Data curation:** Mateusz Sobieski.

**Formal analysis:** Anna Kopszak.

**Investigation:** Mateusz Sobieski, Sylwia Wrona, Maria Magdalena Bujnowska-Fedak.

**Methodology:** Mateusz Sobieski, Anna Kopszak, Sylwia Wrona, Maria Magdalena Bujnowska-Fedak.

**Project administration:** Mateusz Sobieski.

**Resources:** Mateusz Sobieski.

**Supervision:** Mateusz Sobieski, Maria Magdalena Bujnowska-Fedak.

**Validation:** Mateusz Sobieski, Maria Magdalena Bujnowska-Fedak.

**Visualization:** Mateusz Sobieski, Anna Kopszak.

**Writing – original draft:** Mateusz Sobieski, Maria Magdalena Bujnowska-Fedak.

**Writing – review & editing:** Mateusz Sobieski, Anna Kopszak, Sylwia Wrona, Maria Magdalena Bujnowska-Fedak.

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
