## [Decision Letter · Decision Letter 0]

26 Mar 2024

PONE-D-24-05718Screening Accuracy and Age-based Scoring Procedures of the Polish Version of Communication and Symbolic Behavior Scales-Developmental Profile Infant-Toddler ChecklistPLOS ONE

Dear Dr. Sobieski,

Thank you for submitting your manuscript to PLOS ONE. After careful consideration, we feel that it has merit but does not fully meet PLOS ONE’s publication criteria as it currently stands. Therefore, we invite you to submit a revised version of the manuscript that addresses the points raised during the review process.

We look forward to receiving your revised manuscript.

Kind regards,

Amin Nakhostin-Ansari

Academic Editor

PLOS ONE

Journal Requirements:

"Approval from the Bioethics Committee of the Wroclaw Medical University was obtained to conduct the study (number KB – 641/2020). All procedures were performed in accordance with the 1964 Helsinki Declaration and its later amendments."

"The study was financed from the researchers' own funds and the internal funds of Wroclaw Medical University - the project is financed from the scientific grant: "Aspects of preventive care, diagnosis and therapy of patients of different ages under the care of a family doctor, including e-health, telemedicine solutions, coordination and analysis of care effectiveness indicators”; Task SIMPLE number: SUB.C290.19.054. 

Due to obtaining consent from the authors of the original version of the CSBS-DP ITC questionnaire and the general free availability of the questionnaire, the Polish version of the CSBS-DP ITC questionnaire is covered by the CC BY license, with particular emphasis on the original authors and Paul H. Brookes Publishing Co.

The researchers would like to thank Róża Hajkuś and Kamila and Tomasz Chadaj for their help in recruiting study participants."

"The study was financed from the researchers' own funds and the internal funds of Wroclaw Medical University - the project is financed from the scientific grant: "Aspects of preventive care, diagnosis and therapy of patients of different ages under the care of a family doctor, including e-health, telemedicine solutions, coordination and analysis of care effectiveness indicators”; Task SIMPLE number: SUB.C290.19.054. "

4. Thank you for uploading your study's underlying data set. Unfortunately, the repository you have noted in your Data Availability statement does not qualify as an acceptable data repository according to PLOS's standards.

Reviewers' comments:

Reviewer's Responses to Questions

**Comments to the Author**

1. Is the manuscript technically sound, and do the data support the conclusions?

Reviewer #1: Partly

Reviewer #2: Partly

2. Has the statistical analysis been performed appropriately and rigorously? 

Reviewer #1: No

Reviewer #2: Yes

3. Have the authors made all data underlying the findings in their manuscript fully available?

Reviewer #1: Yes

Reviewer #2: Yes

4. Is the manuscript presented in an intelligible fashion and written in standard English?

Reviewer #1: Yes

Reviewer #2: Yes

5. Review Comments to the Author

Reviewer #1: Thank you for inviting me to review this manuscript. The aim of the study was to evaluate the sensitivity and specificity of the Polish version of the CSBS-ITC, to assess the cut-off points and to analyse the predictive validity of method. Participants were 602 children between 6 and 24 months of age. The results showed that the sensitivity of the method varied between .667 and .750, and specificity varied between .854 and .939. Authors conclude that the questionnaire can be used as an effective tool ASD universal screening.

This study has an important goal: to analyse the properties of a screening method. However, the manuscript is rather challenging to read – some parts of the text could be re-organized and the text would benefit from clarifications.

Title: The following words are included in the title “..age-based scoring procedures..” However, the text did not include this kind of information. The title could be modified so that it would better reflect the content of the study.

Abstract: The aims presented in the abstract could be the same as presented in the text before the beginning of the method section. In addition, the results presented in the abstract could follow the aims of the study. Now only the results in terms of the sensitivity and specificity are presented in the abstract. Further, the participants age period could be presented.

Introduction: The text in the introduction is quite well organized. However, the information regarding the CSBS-ITC is somewhat unclear. In the 4th chapter it is presented that the method was adapted to Polish, but in the 5th chapter it is stated that the method was translated. In addition, it is not always clear in the text, whether it is spoken about the original CSBS-ITC or about the Polish version. This could be clarified.

Methods: The text in the Methods section is difficult to follow. A very big portion of the text is about the procedure. The participants are not clearly presented. In addition, abbreviations of the methods are used in the text before the methods are presented. The text could be re-organized under the following sub-titles: Participants, Methods, Procedure. The flow-chart could be used to present how the group of participants was modified.

In Table 1 it is stated that 39,86% of the children had received physical rehabilitation in the past, and roughly 20% of the children had muscle tone or other musculoskeletal system disorders. Still, the inclusion criteria were: living in Poland, speaking Polish as one’s primary language, and giving informed written consent. The proportions presented are very high for the typical population. Please clarify.

Statistical analysis: It is presented that children were divided into age-groups, but the groups are not presented. Further, I did not understand why the psychomotor development was used as a basis for grouping.

Results: The descriptive statistics are not presented at all. In my view it would be important to present them so that the reader has the possibility to reflect the information of the present study in the light of the information presented in previous studies.

Further, the information presented in Table 2 is unclear. Why partial correlations (age controlled) were not used instead?

The results could be presented in the same order as the aims of the study are presented (Data description, Sensitivity and specificity, Cut-off points, Predictive validity).

Discussion: The results are not discussed in the light of the previous findings in the literature, as it should be done in the Discussion section, there are hardly any references mentioned in the text. Due to this, the meaning of the present findings remains unclear. The text in the discussion should be modified so that the present results are discussed in the light of the literature.

Thank you for the possibility to read this study with an important aim. I hope that my comments are useful for the authors.

Reviewer #2: The paper provides detailed evidence about the suitability of the CSBS-DP instrument as a screening tool for ASD. I have a few reservations

1. About half of the participants did not complete the study, although they had previously provided CSBS scores. Most of these cited lack of autism symptoms so these data are clearly not missing-at-random. What were the CSBS scores of the missing group? If they were to be included in the analysis in the no-ASD category, would there be some low CSBS scores (false positives) which would impact on the specificity estimates

2. Please can the authors provide numbers of participants in each age category with an ASD diagnosis. I assume these numbers are quite small.

3. I think there is unnecessary detail in the manuscript, in particular the discussion is far too long which detracts from the main message. For example, regarding the age groups, all that is necessary to say is that, given the age-dependence of CSBS scores, the analysis was appropriately age-stratified. And, while the expected costs methodology is interesting, I think the most commonly used Youden method of obtaining optimal cut-offs is sufficient. This could be quite a short, impactful paper without losing any valuable content.

4. The authors say that correlations between CSBC and ASRS were moderately strong. I'm not sure this is a correct description of a correlation of 0.28 in the 9-12 months age group.

5. I could not understand the scatter plots (only seeing them in grayscale probably doesn't help).

6. PLOS authors have the option to publish the peer review history of their article (what does this mean?). If published, this will include your full peer review and any attached files.

Reviewer #1: No

Reviewer #2: No

---

## [Author Response · Author response to Decision Letter 0]

23 Apr 2024

Response to Journal Requirements:

**The text of the manuscript was adjusted to PLOS One style requirements, as well as file names.**

"Approval from the Bioethics Committee of the Wroclaw Medical University was obtained to conduct the study (number KB – 641/2020). All procedures were performed in accordance with the 1964 Helsinki Declaration and its later amendments."

"The study was financed from the researchers' own funds and the internal funds of Wroclaw Medical University - the project is financed from the scientific grant: "Aspects of preventive care, diagnosis and therapy of patients of different ages under the care of a family doctor, including e-health, telemedicine solutions, coordination and analysis of care effectiveness indicators”; Task SIMPLE number: SUB.C290.19.054. 

Due to obtaining consent from the authors of the original version of the CSBS-DP ITC questionnaire and the general free availability of the questionnaire, the Polish version of the CSBS-DP ITC questionnaire is covered by the CC BY license, with particular emphasis on the original authors and Paul H. Brookes Publishing Co.

The researchers would like to thank Róża Hajkuś and Kamila and Tomasz Chadaj for their help in recruiting study participants."

"The study was financed from the researchers' own funds and the internal funds of Wroclaw Medical University - the project is financed from the scientific grant: "Aspects of preventive care, diagnosis and therapy of patients of different ages under the care of a family doctor, including e-health, telemedicine solutions, coordination and analysis of care effectiveness indicators”; Task SIMPLE number: SUB.C290.19.054. "

**In reference to the two comments above - the role of external funders has been added. The text of the statement is included below and added to the cover letter.**

The study was financed from the researchers' own funds and the internal funds of Wroclaw Medical University - the project is financed from the scientific grant: "Aspects of preventive care, diagnosis and therapy of patients of different ages under the care of a family doctor, including e-health, telemedicine solutions, coordination and analysis of care effectiveness indicators”; Task SIMPLE number: SUB.C290.19.054. The external funders had no role in study design, data collection and analysis, decision to publish, or preparation of the manuscript.

4. Thank you for uploading your study's underlying data set. Unfortunately, the repository you have noted in your Data Availability statement does not qualify as an acceptable data repository according to PLOS's standards.

**The minimal data set was placed on FigShare and, as far as we know, it is generally available for download as an MS Excel xlsx file. **

**The fragment of the manuscript regarding obtaining consent was moved to Methods, information about the written consent obtained and the full text of the EC consent was added.**

Response to Reviewer #1

Reviewer #1: Thank you for inviting me to review this manuscript. The aim of the study was to evaluate the sensitivity and specificity of the Polish version of the CSBS-ITC, to assess the cut-off points and to analyse the predictive validity of method. Participants were 602 children between 6 and 24 months of age. The results showed that the sensitivity of the method varied between .667 and .750, and specificity varied between .854 and .939. Authors conclude that the questionnaire can be used as an effective tool ASD universal screening. This study has an important goal: to analyse the properties of a screening method. However, the manuscript is rather challenging to read – some parts of the text could be re-organized and the text would benefit from clarifications.

**Thank you for your devoted time and attention to reviewing our manuscript. We addressed individual comments (including those regarding the layout of the text) later in the response.**

Title: The following words are included in the title “..age-based scoring procedures..” However, the text did not include this kind of information. The title could be modified so that it would better reflect the content of the study.

**Due to the change in the methodology for determining cut-off points compared to the original version, to a more accurate one in our opinion, we decided to include this in the title. However, we understand that this may lead to confusion, so we have changed the title of the manuscript.**

Abstract: The aims presented in the abstract could be the same as presented in the text before the beginning of the method section. In addition, the results presented in the abstract could follow the aims of the study. Now only the results in terms of the sensitivity and specificity are presented in the abstract. Further, the participants age period could be presented.

**Thank you for your pertinent comments. These were included in the abstract.**

Introduction: The text in the introduction is quite well organized. However, the information regarding the CSBS-ITC is somewhat unclear. In the 4th chapter it is presented that the method was adapted to Polish, but in the 5th chapter it is stated that the method was translated. In addition, it is not always clear in the text, whether it is spoken about the original CSBS-ITC or about the Polish version. This could be clarified.

**As an adaptation, we mean translation, linguistic and cultural adaptation - e.g. matching the phrases asked in the CSBS-DP to those most frequently used by Polish children; matching linguistic phonemes to the development of speech in Polish, etc. We have tried to include appropriate explanations in the text. Moreover, we have marked in the introduction where the original version is mentioned and where the Polish version is mentioned.**

Methods: The text in the Methods section is difficult to follow. A very big portion of the text is about the procedure. The participants are not clearly presented. In addition, abbreviations of the methods are used in the text before the methods are presented. The text could be re-organized under the following sub-titles: Participants, Methods, Procedure. The flow-chart could be used to present how the group of participants was modified.

In Table 1 it is stated that 39,86% of the children had received physical rehabilitation in the past, and roughly 20% of the children had muscle tone or other musculoskeletal system disorders. Still, the inclusion criteria were: living in Poland, speaking Polish as one’s primary language, and giving informed written consent. The proportions presented are very high for the typical population. Please clarify.

**Our goal was to thoroughly explain the methodology used to make the study replicable. We have reorganized this section of the manuscript as suggested. The flowchart is attached to the manuscript as Fig 1. Explanations have been added regarding such a high percentage of children undergoing rehabilitation.**

Statistical analysis: It is presented that children were divided into age-groups, but the groups are not presented. Further, I did not understand why the psychomotor development was used as a basis for grouping.

**Piaget's Sensorimotor Stage theory covers not only motor development (basically, it concerns the development of purposefulness of movements, e.g. pointing, the lack of which is one of the characteristic symptoms of ASD) but also the development of operational and cognitive thinking. We used this division due to the recognized status of this theory and its support in research. We added information about the number of children in subgroups and sex structure in the text.**

Results: The descriptive statistics are not presented at all. In my view it would be important to present them so that the reader has the possibility to reflect the information of the present study in the light of the information presented in previous studies.

**Description and table regarding descriptive statistics have been added.**

Further, the information presented in Table 2 is unclear. Why partial correlations (age controlled) were not used instead?

**Table 2 shows the results of the analysis in subgroups. We performed a separate analysis for each group instead of one analysis on the whole dataset because we wanted to illustrate also how age affects the relationship between the two scales.

Regarding the partial correlation analysis, the rationale behind testing correlations and Age separately and not using partial correlation was that we wanted to check the relationship between ASRS and CSBS, and we suspected a possible relationship between CSBS and Age but not between Age and ASRS. We assumed that Age was only related to CSBS and not to ASRS, because the ASRS was measured at an older age, not the one reported in the variable "Age". The variable Age was the age of computing CSBS and this variable was indeed correlated with the outcome of CSBS. We chose not to use a method assuming a correlation between the age of the CSBS test and the ASRS result because we suspected that any relationship detected between the ASRS scale and age would be likely spurious and bias the results. Additionally, we have examined the standard correlations between ASRS and Age and scatter plots, and no association between these variables was confirmed. To sum up, partial correlation assumes that there is one (or more) confounding variable correlated with two others, but we did not find reasons to assume that this is the case in this study, so we opted not to use the partial correlation coefficient.**

The results could be presented in the same order as the aims of the study are presented (Data description, Sensitivity and specificity, Cut-off points, Predictive validity).

**To maintain the sequence of the order of statistical analyses, the aim has been changed to match the suggestion.**

Discussion: The results are not discussed in the light of the previous findings in the literature, as it should be done in the Discussion section, there are hardly any references mentioned in the text. Due to this, the meaning of the present findings remains unclear. The text in the discussion should be modified so that the present results are discussed in the light of the literature.

**The discussion was expanded to include further potential implications arising from the study and comparison with existing research on CSBS-DP ITC.**

Thank you for the possibility to read this study with an important aim. I hope that my comments are useful for the authors.

**Once again, we would like to thank you for your invaluable comments, the inclusion of which significantly improved the scientific value of our manuscript. We hope that after proofreading, the revised manuscript will receive your approval for publication in PLOS One.**

Response to Reviewer #2: 

The paper provides detailed evidence about the suitability of the CSBS-DP instrument as a screening tool for ASD. I have a few reservations

1. About half of the participants did not complete the study, although they had previously provided CSBS scores. Most of these cited lack of autism symptoms so these data are clearly not missing-at-random. What were the CSBS scores of the missing group? If they were to be included in the analysis in the no-ASD category, would there be some low CSBS scores (false positives) which would impact on the specificity estimates.

**We understand the potential for reduced accuracy of the tool if the full sample is used (including children whose parents did not participate in the full study protocol). On the other hand, a larger number of correctly classified cases could increase the specificity of the tool and thus increase its accuracy.

Nevertheless, we decided to use in the analyzes only the responses from parents who participated in the entire protocol, if only due to the potential denial of the child's symptoms by parents or other possible situations that make diagnosis difficult. 

We believe it is more correct if we use a smaller group of respondents, but subjected to complete evaluation. We considered the population that took part only in first phase of screening for the purpose of prevalence , which in our study was significantly higher than in the official data provided by the Polish National Health Fund, and similar to data from neighboring countries.**

2. Please can the authors provide numbers of participants in each age category with an ASD diagnosis. I assume these numbers are quite small.

**Description and table regarding descriptive statistics have been added (as well with information about numbers of participants with ASD in each age subgroup).**

3. I think there is unnecessary detail in the manuscript, in particular the discussion is far too long which detracts from the main message. For example, regarding the age groups, all that is necessary to say is that, given the age-dependence of CSBS scores, the analysis was appropriately age-stratified. And, while the expected costs methodology is interesting, I think the most commonly used Youden method of obtaining optimal cut-offs is sufficient. This could be quite a short, impactful paper without losing any valuable content.

**Finally, we adapted Youden's approach to calculate ROC and thus determine cut-off points, due to the completely unintuitive results of the misclassification analysis. However, it seems to us that due to the fact that no other scientific article regarding ASD screening tools known to us has used this method, it is worth including a description of this analysis as well. This may be evidence for other scientists that this method is not effective in this case. However, due to the extensiveness of the analyzes and their less usefulness, they were moved to the supplementary file.**

4. The authors say that correlations between CSBC and ASRS were moderately strong. I'm not sure this is a correct description

---

## [Decision Letter · Decision Letter 1]

28 May 2024

PONE-D-24-05718R1Screening Accuracy and Cut-offs of the Polish Version of Communication and Symbolic Behavior Scales-Developmental Profile Infant-Toddler ChecklistPLOS ONE

Dear Dr. Sobieski,

Thank you for submitting your manuscript to PLOS ONE. After careful consideration, we feel that it has merit but does not fully meet PLOS ONE’s publication criteria as it currently stands. Therefore, we invite you to submit a revised version of the manuscript that addresses the points raised during the review process.

We look forward to receiving your revised manuscript.

Kind regards,

Amin Nakhostin-Ansari

Academic Editor

PLOS ONE

Reviewers' comments:

Reviewer's Responses to Questions

**Comments to the Author**

1. If the authors have adequately addressed your comments raised in a previous round of review and you feel that this manuscript is now acceptable for publication, you may indicate that here to bypass the “Comments to the Author” section, enter your conflict of interest statement in the “Confidential to Editor” section, and submit your "Accept" recommendation.

Reviewer #3: (No Response)

Reviewer #4: (No Response)

2. Is the manuscript technically sound, and do the data support the conclusions?

Reviewer #3: Partly

Reviewer #4: Yes

3. Has the statistical analysis been performed appropriately and rigorously? 

Reviewer #3: Yes

Reviewer #4: Yes

4. Have the authors made all data underlying the findings in their manuscript fully available?

Reviewer #3: Yes

Reviewer #4: Yes

5. Is the manuscript presented in an intelligible fashion and written in standard English?

Reviewer #3: Yes

Reviewer #4: Yes

6. Review Comments to the Author

Reviewer #3: The present study aimed to assess the Polish version of CSBS-DP ITC regarding its convergent validity, sensitivity, and specificity. Longitudinally, they also examined the children of different age groups to introduce cut-off scores of CSBS-DP ITC to diagnose ASD. The study is well-written. However, it needs revision. Note that I ask authors to precisely address the changes they make by mentioning the lines of revised text. The answers to the previous reviewers are vaguely addressed.

1- In the background section of the abstract, I suggest naming the assessed tool first. "This study aims to assess the predictive convergent validity OF CSBS-DP ITC ..."

2- The abbreviations are not correctly addressed all over the text. Note that authors should mention the complete form for the first time when using the abbreviation, and, afterward, they are allowed to use the abbreviation alone. The abstract and the full text are considered separate texts in this regard. Please revise the whole draft as I saw mistakes repeatedly.

3- Please be consistent all over the text regarding the name of the attached materials. I refer you to the journal guidelines about how to address them, e.g., S1 File or Supplementary File 2 and Fig or Figure.

4- In the methods section, you have stated "When attempts were made to contact the other parents via e-mail or telephone (N = 849)...". However,=1461-602 is 859. Please clarify this issue in the methods and the first figure.

5- In the methods section, please provide the SD alongside the mean of the age of children at follow-up.

6- Clarify what you meant by "serious health problems" in children.

7- Briefly mention your Spojrzeć w oczy project in the methods section.

8- In the methods section, you said "The ASRS was developed in 2013". I suppose it's wrong (2009), take a look at https://samgoldstein.com/pdf/ASRS-Product-Overview.pdf.

9- In your methods section, it's vague if children were considered autistic when they had positive results of ADOS before your assessment of ADOS or not.

10- In the statistical analysis, mention the details of the software. Moreover, mention the R packages used to perform the analysis, alongside their version.

11- In your text, you said "It was checked whether there were any interactions for the model with the CSBS-DP ITC Total Score explained variable and the Total ASRS and age explanatory variables." while what you did was a mere correlation analysis between the CSBS-DP ITC and age. Please revise the methods in this regard.

12- In the results section, please do not write multiple call-outs for the figures, e.g., "Results are presented in Table 3 and Figs 2 and 3" and "The results are shown in Figure 3."

13- In the results section, you said that "the highest sensitivity and specificity values were achieved using the Youden method and cut-offs did not change depending on the assumed probability of ASD occurrence." However, you only provided one table of Youden results. Please clarify the exact probability you considered to achieve the sensitivity and specificity values in Table 4.

14- The discussion is too long and excessively focused on the study methods. I suggest shortening the discussion by omitting paragraphs starting with "differences between the prevalence" and "Due to the small number of cases".

15- The S1 and S2 files are the same! Revise your files as you should have provided the Polish version in the S1 file!

Reviewer #4: It was a pleasure to review your work. You have conducted an excellent study, and I commend you for addressing and correcting all the issues raised by previous reviewers. I believe your paper is nearly ready for acceptance. I have just two minor recommendations:

1. I recommend providing explanations for abbreviations in the footnotes of each table to ensure they can be understood individually.

2. I believe the discussion section could be more concise to deliver the study's key points more directly and efficiently. However, since there is no word count limit in the PLOS ONE journal, this is merely a recommendation.

7. PLOS authors have the option to publish the peer review history of their article (what does this mean?). If published, this will include your full peer review and any attached files.

Reviewer #3: No

Reviewer #4: **Yes: **Fateme TaghaviZanjani

---

## [Author Response · Author response to Decision Letter 1]

5 Jun 2024

Reviewer #3: The present study aimed to assess the Polish version of CSBS-DP ITC regarding its convergent validity, sensitivity, and specificity. Longitudinally, they also examined the children of different age groups to introduce cut-off scores of CSBS-DP ITC to diagnose ASD. The study is well-written. However, it needs revision. Note that I ask authors to precisely address the changes they make by mentioning the lines of revised text. The answers to the previous reviewers are vaguely addressed.

Thank you for taking the time to revise our manuscript and for your valuable and relevant comments. We will refer to them in the following points.

1- In the background section of the abstract, I suggest naming the assessed tool first. "This study aims to assess the predictive convergent validity OF CSBS-DP ITC ..."

Response: We have corrected this fragment of the abstract.

2- The abbreviations are not correctly addressed all over the text. Note that authors should mention the complete form for the first time when using the abbreviation, and, afterward, they are allowed to use the abbreviation alone. The abstract and the full text are considered separate texts in this regard. Please revise the whole draft as I saw mistakes repeatedly.

Response: We reviewed the manuscript again, correcting any inaccuracies regarding abbreviations.

3- Please be consistent all over the text regarding the name of the attached materials. I refer you to the journal guidelines about how to address them, e.g., S1 File or Supplementary File 2 and Fig or Figure.

Response: We have unified the naming of figures and additional files.

4- In the methods section, you have stated "When attempts were made to contact the other parents via e-mail or telephone (N = 849)...". However,=1461-602 is 859. Please clarify this issue in the methods and the first figure.

Response: We made an error in calculating the number of parents we were unable to contact. We have corrected this in both the text and Fig 1.

5- In the methods section, please provide the SD alongside the mean of the age of children at follow-up.

Response: We added information about the standard deviation of children's age at follow-up.

6- Clarify what you meant by "serious health problems" in children.

Response: We have clarified this issue in the footnote of the table 1 regarding sociodemographic characteristics.

7- Briefly mention your Spojrzeć w oczy project in the methods section.

Response: Information about the project's goals and a reference to additional research results have been added

8- In the methods section, you said "The ASRS was developed in 2013". I suppose it's wrong (2009), take a look at https://samgoldstein.com/pdf/ASRS-Product-Overview.pdf.

Response: We took the information regarding 2013 as the year of development of ASRS from the manual for the Polish version of ASRS. We have corrected this information.

9- In your methods section, it's vague if children were considered autistic when they had positive results of ADOS before your assessment of ADOS or not.

Response: We have tried to clarify this issue by describing it in more detail in the Procedures.

10- In the statistical analysis, mention the details of the software. Moreover, mention the R packages used to perform the analysis, alongside their version.

Response: We added information about the used versions and packages.

11- In your text, you said "It was checked whether there were any interactions for the model with the CSBS-DP ITC Total Score explained variable and the Total ASRS and age explanatory variables." while what you did was a mere correlation analysis between the CSBS-DP ITC and age. Please revise the methods in this regard.

Response: Due to the lack of fulfillment of the assumption about the linear relationship of the studied variables, it is not possible, for example, to create a linear model with interactions. However, in fact, this sentence may be viewed as inaccurate and implying stronger significance than we have actually demonstrated. We have corrected this sentence.

12- In the results section, please do not write multiple call-outs for the figures, e.g., "Results are presented in Table 3 and Figs 2 and 3" and "The results are shown in Figure 3."

Response: We have removed call-outs to tables and figures except those that we consider necessary to understand the text of the work and maintain an appropriate flow for the reader.

13- In the results section, you said that "the highest sensitivity and specificity values were achieved using the Youden method and cut-offs did not change depending on the assumed probability of ASD occurrence." However, you only provided one table of Youden results. Please clarify the exact probability you considered to achieve the sensitivity and specificity values in Table 4.

Response: We added information about the considered prevalences of ASD for performing Youden method ROC analysis. 

14- The discussion is too long and excessively focused on the study methods. I suggest shortening the discussion by omitting paragraphs starting with "differences between the prevalence" and "Due to the small number of cases".

Response: We have removed the cited paragraphs to shorten the discussion.

15- The S1 and S2 files are the same! Revise your files as you should have provided the Polish version in the S1 file!

Response: Due to a mistake in adapting file names to PLOS One guidelines, the same file was sent twice. We have corrected the error.

Thank you again for all your comments. We hope that after taking them into account, our manuscript has improved in quality and is suitable for publication in PLOS One.

Reviewer #4: It was a pleasure to review your work. You have conducted an excellent study, and I commend you for addressing and correcting all the issues raised by previous reviewers. I believe your paper is nearly ready for acceptance. I have just two minor recommendations:

Thank you very much for such a positive opinion about our manuscript. We will refer to them in the following points.

1. I recommend providing explanations for abbreviations in the footnotes of each table to ensure they can be understood individually.

Response: We provided explanations for abbreviations in the footnotes.

2. I believe the discussion section could be more concise to deliver the study's key points more directly and efficiently. However, since there is no word count limit in the PLOS ONE journal, this is merely a recommendation.

Response: We have shortened the discussion slightly as suggested.

Thank you again for spending your valuable time reviewing our manuscript.

---

## [Editor Report · Decision Letter 2]

7 Jun 2024

Screening Accuracy and Cut-offs of the Polish Version of Communication and Symbolic Behavior Scales-Developmental Profile Infant-Toddler Checklist

PONE-D-24-05718R2

Dear Dr. Sobieski,

We’re pleased to inform you that your manuscript has been judged scientifically suitable for publication and will be formally accepted for publication once it meets all outstanding technical requirements.

Kind regards,

Amin Nakhostin-Ansari

Academic Editor

PLOS ONE